# A Bibliometric Analysis and Review of Nudge Research Using VOSviewer

**DOI:** 10.3390/bs13010019

**Published:** 2022-12-25

**Authors:** Chenjin Jia, Hasrina Mustafa

**Affiliations:** School of Communication, Universiti Sains Malaysia, Gelugor 11800, Malaysia

**Keywords:** nudge, choice architecture, bibliometric analysis, Web of Science, VOSviewer

## Abstract

With growing demands of decision making in the current era, the impact of the drivers behind individuals’ preferences and institutional strategies becomes prominent. Coined in 2008, *nudge* is used to describe incentives for individuals’ choices with foreseeable outcomes but without exclusion of alternative choices or reliance on financial stimuli. Consequently, nudge and its application in real-world situations led to a prosperous surge of studies in multiple disciplines. However, we are still facing a dearth of in-depth understanding of the status quo and future directions of research on nudge in a comprehensive fashion. To address the gap in knowledge, the present study adopted a bibliometric analysis of the existing literature related to the investigation and application of nudge by analyzing 1706 publications retrieved from Web of Science. The results indicated that (a) being a relatively newly developed theory, interest in nudge in academia has expanded both in volume and disciplines, with Western scholars and behavioral economists as the backbones; (b) future studies in nudge-related fields are expected to consolidate its current frontiers in individual behaviors while shedding light on new territories such as the digitalized environment. By incorporating state-of-the-art technologies to investigate extant research, the present study would be pivotal for the holistic understanding of the studies on nudge in recent years. Nevertheless, the inclusiveness and comprehensiveness of the review were limited by the size of the selected literature.

## 1. Introduction

We are faced with decision making every day, from the formulation of state policies and development strategies of companies to individuals’ preferences in provision, apparel, and accommodation. However, many times, individuals have limited rationality in the decision-making process. Available information, social influences, and intuition prevail and guide individuals’ decisions [1]. Therefore, behavioral economics research has shown application in helping people make better decisions, effectively helping governments and organizations of all kinds develop and implement public policies that better serve individuals. “Nudge” is a result of the rise of behavioral economics. Behavioral economics does not ban any options, limit freedom of choice, use economic leverage, or give orders. Instead, it changes people’s choices or economic behavior in a predictable way by changing the way they make decisions. 

The concept of nudge comes from the book *Nudge* co-authored by behavioral economist Richard Thaler and law professor Cass R. Sunstein. They define the concept of nudge as “any aspect of the choice architecture that alters people’s behavior in a predictable way without forbidding any options or significantly changing their economic incentives” [2]. They argued that to be considered a mere nudge, the intervention had to be accessible and cheap, an example of which is the design of the school canteen mentioned in the book. To make students’ diets healthier, canteen managers place healthier foods, such as vegetables and fruits, in prominent, easy-to-choose locations, while moving unhealthy junk foods to unobtrusive, hard-to-choose corners. Unlike rationally convincing students of the importance of healthy eating, this approach uses students’ inertia to make their diets healthier [2]. From a practical point of view, this approach is more direct and effective than rational persuasion.

Why is nudge necessary for human behavioral decisions? Thaler and Sunstein pointed out that the daily decision-making behavior of the public is not always absolutely rational, as economists believe, but a “social human” paradigm with shortcuts and cognitive biases [3]. The irrational behavior of individuals can be perceived in four dimensions: cognition, emotion, willingness, and action. At the cognitive level, human judgment and decision making usually involve two main cognitive systems: the intuition-based heuristic system (System 1) and the rationality-based analytical system (System 2) [4]. The interaction of System 1 and System 2 constitutes the human mindset. This division of labor works well in most cases because System 1 is usually very good at perceiving the world around it, its familiarity models are accurate, and its short-term predictions are usually accurate. Precisely because System 1 is unconscious, the resulting cognitive biases are difficult to self-perceive, and if System 2 incorrectly accepts these cognitive illusions, it is difficult to avoid making the wrong decisions. At the emotional level, people tend to be unrealistically optimistic that they are the “lucky ones” and consider avoiding loss more than seeking profit. At the level of willingness and action, humans tend to have status quo bias and lack self-control to resist temptation; at the same time, they are easily influenced by the environment and make unwise decisions under the effect of social norms, herd mentality, and peer pressure.

As a tool of intervention in human choice, nudge is similar to the design of Persuasive Technology (PT), an interactive technique that can change people’s attitudes and behaviors proposed by Professor Fogg [5]. Fogg believes that the prerequisite for performing desired behavior is the person’s motivation, abilities, and triggers. These three factors must be present simultaneously for the changed behavior to occur. When motivation is insufficient, incentive to enhance motivation should be set up; when ability is insufficient, guides to enhance ability should be designed; when both motivation and ability are satisfied, we need to create some kind of reminder to trigger the person’s behavior [6]. The goal of both PT design and nudge is to influence individuals’ behavioral decisions. However, Persuasive Design aims at linking attitude and behavior change [7] and can be referred to as an attitude-oriented design strategy, whereas nudge is directly related to decision making [8], a decision-oriented design strategy. Additionally, PT uses a rational persuasion approach, while nudge is based on the irrational, unconscious elements of human behavior.

Nudge has led to a revolution in behavioral science research and has received widespread attention from scholars in management, economics, psychology, medicine, education, etc. It has been applied to various fields as a method to influence people’s behavioral decisions. Examples include influencing consumers’ dietary decisions [9,10], encouraging investors to implement sustainable and responsible investment decisions [11,12], promoting students’ self-directed learning educational decisions [13,14], reducing social media users’ misinformation sharing decisions [15,16,17], and changing individuals’ pro-social behavior decisions [18,19,20].

However, nudge is not always effective. Because nudge is applied by different mechanisms and to different populations, results from some studies have found limited effects of nudge [18,20,21,22]. A quantitative analysis by Hummel [23] showed that only 62% of nudges were statistically significant. Furthermore, a recent preprint tested the effectiveness of nudge in improving attitudes toward shared electric scooters. The results indicated that nudge was not only ineffective, but also worsened attitudes toward shared electric scooters and reduced the expected reverse effect [24].

It is worth noting that some scholars have also discussed the ethical issues of choice architecture in nudge [25,26,27]. The core of nudge has always been to influence (or manipulate) human behavior, especially digital nudging, and since there is no neutral way to present choices, all decisions related to user interface design can influence user behavior [28]. So, choice architects need to consider moral and ethical issues when designing nudge. For example, Lembcke et al. [26] discussed how much effort is reasonable for individuals to put into protecting their freedom of choice, how much concealment can be tolerated and still be considered transparent, or how the goals of the choice architect need to be aligned with the individual’s goals for the nudge to be considered reasonable. It is clear that nudge is intended to pursue the public interest and social welfare, yet there are still many uncertain conditions and methods to be explored, especially how to guarantee that the architects of choice have good design intentions so that nudge will not be abused and achieve better behavioral decision making.

Overall, we understand that nudge takes liberal paternalism as its spiritual core and uses different choice architecture mechanisms to advance the goal of boosting, both by providing options for human freedom of choice and by increasing the chances of making better choices. In recent years, behavioral science researchers have been exploring the potential applications and research of nudge in various fields while using different methods to obtain more reliable and generalizable evidence. The existing reviews on nudge are basically systematic literature reviews, with scholars focusing on topics such as healthy diet [29,30], medical care [31], and online user behavior [32,33].

Existing reviews, systematic reviews, and meta-analyses on nudge are becoming increasingly common. The systematic review is based on synthesizing primary research evidence to provide up-to-date knowledge for nudge’s research by addressing specific research questions. Meta-analysis is a quantitative-based systematic evaluation that combines the results of many empirical studies to assess the effectiveness of different nudge mechanisms for behavioral interventions through appropriate statistical methods. However, the available reviews tend to focus on a single aspect of the current status [32,34], and rarely provide a macroscopic view of the development process and trends in present nudge research. Bibliometrics can assist researchers in better understanding the large number of publications, provide visualization to help researchers identify which fields have achieved significant outcomes, and map future research directions based on this information [35]. Our work in this paper could contribute to fill the gap in this field.

Though the investigation of nudge has become a popular and emerging research topic, we are still facing a paucity of analysis on the relationship between the structure, evolution, collaboration of existing literature, and the clarification of potential research directions. So, this paper adopts a bibliometric analysis of nudge to understand and explore the current state of research on the application of nudge to individual behavior and organizational and governmental decision making. The findings of this study will help behavioral scientists, researchers, decision makers, and institutions of higher education identify research hotspots and emerging trends in nudge and will inform their future research efforts. Specifically, the following research questions would be answered in the present study:RQ1: What trend in publication quantities and national and organizational involvement could be identified from nudge research during the period of 2012–2022?RQ2: What are the most dominant nudge research disciplines and thematic clusters?RQ3: What are the potential areas and future directions of nudge research?

## 2. Materials and Methods

With the development of technology and continuous investment in scientific research, bibliometric analysis has been applied to remedy the limitation of conventional narrative literature review in the evaluation of academic contribution, assessment of merits of studies, and determination of research trends for literature of growing quantity. Bibliometric analysis is an extensive and accurate method for examining and analyzing large amounts of scientific data. The technique aims to understand the interconnections between journal citations and summarize updates on current or rising research topics [36]. As the availability and operability of bibliometric software and scientific databases have increased, this quantitative analysis of the literature, independent of personal subjectivity and other non-scientific factors, is widely used in multiple disciplines.

We first chose the Web of Science (WoS) for bibliographic research on nudge, which is a web-based product developed by Clarivate Analytics and includes the three major citation indexing databases (i.e., SCI, SSCI, A&HCI). WoS includes authoritative and influential journals in various subject areas, and its strict selection criteria and citation indexing mechanism make it one of the most important basic evaluation tools in bibliometrics and scientometrics, while serving as a literature search tool [37]. Previous researchers have argued that WoS has a significant advantage over other databases because the journals it includes demonstrate a high level of editorial rigor and the best practices [38,39]. To quantify the bibliographic material in nudge-related studies, we selected the WoS Core Collection database and set up the following search profile: TS = (topic: “nudge”) AND (title: “nudge” OR “choice architecture”). The search was conducted in late August 2022 and was limited to documents published between 2012 and 2022. The initial search yielded a list of 1752 publications. Afterwards, we manually excluded publications not related to nudge by browsing titles and abstracts, such as “nudged elastic band” used in the discussion of chemical materials. According to this criterion, 1706 references were obtained. We extracted these publication data text files containing useful information about each publication, such as category, journal name, country, organization, and keywords, anchored as the basis for the bibliometric analysis in this work, and enabling us to answer the research questions more explicitly.

There are many tools available for bibliometric analysis, such as CiteSpace, VOSviewer, and HistCite, which provide visual views based on user interfaces, the Bibliometrix package in R, which is based on code commands, and Pajek and Gephi, which focus on constructing complicated network analysis. Among them, Visualization of Similarities viewer (VOS) [40] is becoming increasingly popular in bibliometric studies, with its outstanding visualization capabilities and usability to load and export information from many sources for creating maps based on network data, and to visualize and explore these maps [37]. Citation links, bibliographic coupling, and co-occurrence analysis allow researchers to obtain the themes or clusters used in the titles and abstracts of countries, institutions, and published papers [41]. The software is widely used in bibliometric studies for analysis in different fields such as geography [42], agriculture [43], knowledge management [39], and education [44]. Therefore, in this study, VOSviewer (version 1.6.18) software is used as the main metric tool to visualize and analyze the key hotspots and development evolution of nudge research in the form of knowledge graphs with Network Visualization, Overlay Visualization, and Density Visualization presenting its keyword co-occurrence and literature co-citations.

To generate visualization results for the bibliographic analysis, we imported the downloaded data files into VOSviewer. It allows us to select and set parameters according to different analysis purposes and data sources, as it is usually necessary to perform data cleaning when creating maps based on web data. Therefore, the parameters of this study are set as follows. (a) When creating mappings based on text data, it is possible to use the thesaurus files provided by VOSviewer to merge or ignore certain terms. Therefore, “nudge”, “nudging”, and “nudges” were merged using the thesaurus file, and keywords not relevant to this study that were not screened out manually, such as “nudged elastic band”, were ignored for more accurate clustering analysis. (b) The association strength method was chosen as the strength of association between normative items [45], which was considered to be the most consistent with the normalized method for this study. (c) After testing, the layout with the parameter of attraction set to 2 and the parameter of repulsion set to −1 (creating a map of the author’s co-authorship network) and 0 (creating a map of the co-occurrence network of keywords or a map of the citation network of documents) yielded the optimal visual outcomes [46]. In addition, the other options are default parameters.

Based on the above criteria for searching, filtering, and data processing, Figure 1 presents the bibliometric flow implemented in this study.

## 3. Results and Discussion

### 3.1. Yearly Publication, Document Type, and Research Categories

The number of publications on nudge research is presented in Figure 2, which depicts the development from year 2012 to 2021. The overall growth trend is supported by an increasing number of published articles. In terms of the average annual number of publications, the observed trend in research can be divided into three phases. (1) Before 2015, it was a slow growth period for nudge research, with the number of publications remaining below 100 per year. The average number of publications was about 76 per year. (2) In 2016–2017, the number of publications experienced a slight decline. To explore the reasons for the decrease in publication, we compared the publication categories between the two years. The comparison found that research in economics, law, and ethics were the categories that attracted the most attention, with 20 more publications in these three categories in 2016 than in 2017, while publications in 2017 were more dispersed across disciplines. (3) Research in nudge has grown rapidly since 2018, with a slight negative growth in publications in 2020 likely due to COVID-19, with the number of publications peaking in 2021 (n = 315, 18.46%), and a further increase in research trends likely to be seen in 2022.

We examined the types of documents that included nudge studies, as shown in Table 1. Publications with titles of nudge and nudge topics are indicated by document type. The statistics shown indicate that 71.34% (n = 1217) of the documents were concentrated in one category and were published as articles. Editorial material accounts for 8.50% of the document types, probably because nudge, while being considered an emerging research topic that has received a lot of attention from scholars, has also been discussed and disputed in informal settings. Conference papers accounted for 7.5% of the documents. Because nudge has also been proposed from 2008 to the present in no more than two thousand papers and the findings are not entirely clear, reviews account for only 3.81% of publications. Other document types include book chapters, conference abstracts, news, notes, book reviews, and letters.

Data for the research categories were generated from the search results of the WoS database. Table 2 shows the top 20 research categories of nudge publications. Nudge originated from behavioral economics, and Thaler, the founder of behavioral economics, pioneered the introduction of psychology into economic research, focusing on human behavior, especially human economic behavior. It was later applied to various categories as a tool to intervene in human decision making. As such, the statistics show that nudge research consists of a wide variety of disciplines, with “Economics” (n = 226, 13.25%) remaining the most studied discipline in nudge. “Public Environmental Occupational Health” (n = 116, 6.8%) and “Ethics” (n = 102, 5.98%) were also important research categories for nudge. The categories “Psychology Multidisciplinary”, “Political Science”, “Public Administration”, and “Law”, as the first areas of applied research in nudge, continue to receive attention. In addition, nudge-related research covers such categories as “Social Science Biomedicine”, “Business”, “Environmental Science”, “Nutritional Dietetics”, and other categories, aiming to focus on better serving people in their daily lives and guiding them to choose healthier and more sustainable decisions. Notably, an emerging trend in applying nudge in computer-related fields such as “Computer Science Information Systems” and communication is witnessed, especially after the introduction of digital nudging by Weinmann et al. [47].

#### 3.1.1. Journal Distribution

In general, Nudge Theory can be applied in research pertaining to decision making. We found that articles on nudge-related research are published in a wide range of journals, indicating a significant development in the field. The 1706 articles screened and selected by the researchers were published in a total of 1006 different journals. The top 20 most productive journals are summarized in Table 3 and their number of citations and the impact factors of the journals are reported. It should be noted that we found two journals (i.e., *Journal of Chemical Physics* and *Journal of Chemical Theory and Computation*) from the data obtained from the WoS database, whose keywords are “nudged elastic band”, which is a technique for finding transition paths between a given initial state and final state in chemical research. It does not match the original meaning of “nudge” and is hence excluded. Two books and conference proceedings were also excluded, and the final top 20 journals were ranked according to their number of publications. In the case of a tie, the impact factor of the journal was taken into account.

When journals are ranked based on the number of published articles, their citation counts do not correspond to their rankings. Despite publishing a small number of papers, several journals have relatively high citation counts. To explore the reasons for this, we created a treemap of the average citations (AC) of the top 20 journals (as shown in Figure 3), which is an efficient way to visualize the data in a space-saving manner [48]. The data in Table 3 show that six of the most productive journals are related to medicine and health, while three of the top five journals with the highest AC in Figure 3 are also related to health behavior and health consumption. Although nudge originated from behavioral economics, economics journals are not the most productive. This is presumably because in medicine and health, which have more to do with individual behavior than economics and management, the design and interventions of nudge are more operational. Researchers hope to improve personal health or promote public health by slightly intervening in people’s behavior, for example, by encouraging people to eat more healthily or to have regular health checkups.

Specifically, the most published journal, *American Journal of Bioethics*, with a total of 33 nudge-related studies and 483 citations, is with a modest citation count per paper (n = 13.7). The main reason is that the paper by Blumenthal-Barby and Burroughs [49] has been cited 194 times in WoS. The journal with the highest total and average number of citations was *BMC Public Health*, with studies such as Hollands et al. [50] and Arno and Thomas [51] contributing the major citations. *Nature*, one of the most prestigious scientific journals in the world, published nine papers with only 122 citations, five of which were related to vaccination against COVID-19. While *Review of Philosophy and Psychology* published only 11 papers that focused on nudge in political and legal applications, the number of citations was 295, ranking third in average citations (n = 26.8). In addition, *Food Quality and Preference*, a journal related to sensory science and food research that primarily focuses on the use of nudge to promote healthier food choices or healthier eating habits among consumers, also contributed the third highest number of citations (n = 324) with an average of 24.9 citations.

We used VOSviewer to analyze 1706 articles for co-citation and formed a journal co-citation network with four clusters containing 332 journals, each with a minimum number of citations of 30. As can be seen in Figure 4, each node represents a journal, with its size indicating the number of papers published, and the lines between the nodes indicate the intensity of co-citation, with thicker lines representing higher intensity.

The most visible cluster in the co-citation network is red, with 94 nodes. Among them, *American Economic Review*, with leading impact factors in its field, stands as the central position of the red cluster with the highest intensity of co-citations. The blue cluster has 77 nodes, and the most prominent one is *Science*, which has co-citation links to the other three clusters, although it has published only five nudge-related papers. *Psychological Science* and *Journal of Personality and Social Psychology*, also in the blue cluster, are also important, as they both belong to the Association for Psychological Science (APS) and focus on the frontiers and applications of psychology, and their co-citation relationships are relatively strong. The yellow cluster contains 76 nodes and the journals in this cluster are mostly related to health and diet. The highest co-citation is Appetile, which focuses on the behavior of humans and nonhuman animals toward food and has made important contributions to the study of using nudge to guide consumers to make healthy dietary choices. The most distinguished one out of the 85 nodes within the green clusters is the book *Nudge, nudge, think, think*, by John et al. [52] based on *Nudge* [2]. While acknowledging the power of nudge, they argue that a particular democratic institutional framework is needed to provide an environment that evokes public thinking that promotes listening and reasoned argument among citizens, as well as the type of reflection that can lead to shifts in preferences. Other journals in the green cluster focus on ethics, law, and behavioral studies, all of which have played important supporting roles in nudge research.

#### 3.1.2. Country and Institution Distribution

In bibliometric analysis, country and institution, as important analytical variables, can reflect the research intensity and contribution of different regions or institutions in the research field. By analyzing the citation and co-citation of publications from different countries or institutions, we can gauge their academic level and collaborative networks [53].

From the data obtained from the WoS database, we found that the 1706 publications were distributed among 79 countries, and Table 4 shows the top 10 countries with the highest number of publications, which accounted for 90.7% of the total number of nudge publications (n = 1548). The United States ranked first with 631 publications, accounting for 37.0% of all publications, far ahead of other countries. England stands as the runner-up (260/1706, 15.2%), followed by Germany (159/1706, 9.3%).

Next, we analyzed the most influential countries for nudge research through bibliographic coupling links. The logic behind bibliographic coupling is that two texts with a high number of shared literature references will be similar in content [54]. This means that the coupling analysis shows the number of identical references cited by the documents as a measure of the collaboration of the country to which the publication belongs. We selected the analysis type “bibliographic coupling” in VOSviewer, and the unit of analysis was “countries”. Additionally, we selected the minimum number of documents for countries to be 1 in VOSviewer to obtain the maximum number of links generated between countries. The studies of Ukraine and Iraq among the 79 countries in the network were not interlinked with other countries, so the maximum number of linked items in the final generated coupling network was 77 countries.

The country analysis by bibliographic coupling is presented in a network visualization of the five main clusters, as shown in Figure 5. The most striking one of the clusters is undoubtedly the blue cluster, represented by the United States. It is the first one to apply nudge in practice, not only because the authors of *Nudge*, Thaler and Sunstein, were an American economist and an American legist, but also for the size of the country, the number of research scholars, and the investment in scientific research. During his term in office, former U.S. President Barack Obama signed an executive order establishing the Social and Behavioral Sciences Team, who translated Nudge Theory into improvements in federal policies and programs with success [55]. Canada, South Africa, Thailand, and the Philippines are also in this cluster, indicating that these countries cite similar research articles in nudge research.

The green cluster contains 23 countries, most notably England, which was also the earliest country to start research and application of nudge. The rest of the cluster is dominated by European countries (e.g., France, Switzerland, Belgium, Portugal, etc.), Asian countries (e.g., Vietnam, Korea), and Oceanic countries (e.g., New Zealand). These countries have cited similar articles in their nudge studies. Twenty-eight countries in the red cluster are spread out, ranging from European countries such as Germany, Italy, Sweden, and Norway, to Asian countries such as China, India, and Japan, and American countries such as Brazil and the Dominican Republic. In addition to being geographically distant, these countries are cited in similar articles. Regarding the relatively sparser clusters, Australia and Denmark are most visible in the purple and yellow clusters, respectively.

To gain insight into which countries have recently embarked on the nudge study actively, we created an overlay visualization of the country analysis. The score values are color mapped by taking the average year of country studies by default, as shown in Figure 6. We found that the average year of active research on nudge across countries began in 2017, which indicates that within the decade when nudge was first presented, its effectiveness and academic potential for behavioral interventions was far underestimated. Since 2017, the United States, U.K., France, Denmark, and Canada have taken the lead in starting or expanding participation in nudge study, followed by countries such as Italy, Germany, The Netherlands, China, and Japan. In the past two years, more and more developing countries have also invested in nudge research (e.g., Indonesia, Thailand, Philippines, Nigeria, etc.). We believe that this phenomenon is inseparable from the level of economic development and academic research in a country. Nudge was proposed to make better decisions for people’s health, wealth, and happiness, and countries with high levels of development focused on people’s well-being earlier and started research on nudge earlier. Therefore, based on the analysis and understanding of the status, we can expect and predict a vigorous development of research and applications in related fields.

#### 3.1.3. Author Distribution

According to Thaler [56], the emerging and interdisciplinary field of behavioral economics allows scholars to understand human behavior from a more humanitarian perspective. By the same token, nudge-oriented research has become attractive to a growing number of researchers. Consequently, to obtain a synopsis of nudge-related studies, one of the most pivotal tasks is to identify the most productive and influential authors in the field. We conducted an author citation analysis to identify the top 10 most productive authors and rank them by document and citation. Table 5 shows the results of this analysis. Noticeably, that the number of publications is a metric that should be analyzed with discretion, taking into account factors including the length of each paper, the quality of the journal, and the number of authors per work [57]. The table is sorted by the number of articles by author, and in case of ties, the citations per author were considered. In addition, the h-index, a composite indicator combining productivity and impact, was appended to the table.

Among the top 10 authors, Cass R. Sunstein is the most productive, with 20 publications, and he also ranks first in citations (n = 1057). Given his groundbreaking contribution in relevant fields (e.g., co-authored the book *Nudge*, founded the Behavioral Economics and Public Policy Program at Harvard Law School), his leading position in both publication and citation numbers is understandable and well-expected. He has worked closely with the U.S. Behavioral Insights Team since its inception. In 2020, the World Health Organization appointed him as chair of its technical advisory group on Behavioral Insights and Sciences for Health. His work and research laid the foundations of behavioral economics and provided the shoulders of giants for subsequent scholars. The runner-up in the author list is Peter John, a professor in the Department of Political Economy at King’s College London, with a total of 15 nudge research publications. He is adept at using randomized controlled experiments to explore how nudge can be applied in public policy, and how best to engage citizens interested in public policy and management, and in turn deploy behavioral interventions.

Notably, four of the top ten productive scholars have medical backgrounds. Mitesh S. Patel, Anne Thorndike, Douglas Levy, and Joline Beulens all have a medical and health perspective on interventions that use behavioral economics strategies to improve individuals’ dietary intake and health behaviors. Of those, Mitesh S. Patel has 14 publications, and his research focuses on integrating nudge with scalable technology platforms to improve health and healthcare. He has collaborated with health systems, insurers, employers, and community organizations to conduct clinical trials using nudge, such as digital health interventions using wearables and smartphones, and health system interventions using electronic health records, advancing the research and application of nudge. The most cited of this cohort of researchers is Anne Thorndike, with 10 publications and 465 citations. Her research concentrates on the use of nudge and choice architecture, such as traffic light labels [58], to guide people to healthier food choices and maintain healthy lifestyles.

To further explore the authors’ collaborative research, we conducted a co-authorship analysis, which is a tool used to identify key organizations and scientists and examine their associations [59]. VOSviewer identified 4557 authors based on the WoS data, and the calculation generated 600 items when the minimum co-authorship of articles was set to two. The largest set of connected items contains 32 items, and Figure 7 shows the full co-authorship visualization. Interestingly, while Cass R. Sunstein was the most influential author, the highest scoring co-authorship was Denise de Ridder. She has 21 links, which implies that she has collaborated with 21 authors. Denise de Ridder is a professor of psychology at the Department of Social, Health and Organizational Psychology at Utrecht University and project leader of various research projects in the field of self-regulation and facilitation. She shows that she is proficient in collaborating with colleagues whose nudge research focuses on exploring self-awareness in nudge implementation and how nudge can help people to make healthier food choices. Besides her, the previously discussed authors are also relatively conspicuous in the co-authorship analysis visualization, with an average number of links of 10, which is inseparable from their overall number of publications. In conclusion, no scholars are in a dominant position in nudge research, as the field is still in its infancy and researchers can enhance their collaboration to explore more the potential of nudge in various fields of research. This might be an advantage for potential researchers and ongoing studies, as journal editors prefer a small group of highly productive researchers when deciding which articles to publish [60].

#### 3.1.4. Keywords Co-Occurrence Analysis

The last research questions for this study to explore concerned existing or future relationships between themes in the nudge studies by focusing on the content of the publication itself. For this purpose, keyword co-occurrence analysis was employed, which is a technique for examining the content of the publication by extracting keywords from the full text of the publications. Applied longitudinally, keyword co-occurrence analysis can be used to predict future research in the field with a view to enriching the study’s interpretation of co-citation analysis (in the past) or bibliographic coupling (in the present) and predicting the development of the field (in the future) [35]. In addition, to obtain more accurate results, less relevant keywords were manually removed and a minimum number of occurrences of keywords of six was set as a threshold level. The network visualization was constructed based on the co-occurrence frequency of 294 keywords out of a total of 5444 retrieved keywords, as shown in Figure 8.

In network visualization, each keyword is represented by a node, and the size of each node represents the number of publications in which that keyword appears. The clusters of the nodes are reflected by corresponding colors, with the distance between various clusters indicating the relatedness between them. Specifically, a close relatedness between two clusters could be identified if the distances between them is shorter, and vice versa. There are four main clusters in the network visualization shown in Figure 8, which are red, green, blue, and yellow.

First, red clustering includes topics related to behavioral economic theory, public policy making, and ethical discussions. As an innovative approach to address policy issues, nudge is becoming increasingly popular in the field of public administration. In a recent study, John et al. [61] designed randomized trials of support for nudge and deliberate nudge in response to top-down regulation and freedom of choice. The results of the experiment showed that public support for both nudge policy options is higher compared to top-down regulation. They also found that support for nudge and deliberate nudge is more correlated with perceived fairness than with perceived efficacy. Similarly, a study showed that nudge interventions positively moderated the impact of two-way risk communication on public value consensus [62], which suggested that nudge can play a better role in public management than injunctive interventions. From another perspective, some studies have focused on whether there are ethical issues with nudge, such as doubting whether nudge may have the undesirable consequences of manipulating choice, reducing autonomy, and unintended behavior [63,64]. Conversely, others have assessed and argued that nudge does not usually interact with people’s rationality in a problematic way [65], and that ethicists should remain open to its application [66].

The green cluster focuses on online information, social media, digital nudging, and their impacts. Compared with traditional offline contexts, decision making in digital contexts is more dependent on human–computer interaction interfaces, so the interface design of human–computer interaction can have a significant impact on the decision-making process. This influence includes two main aspects: the interface provides the necessary elements for decision makers to access relevant information, and the way the interface provides this information affects the cognitive process, producing different decision outcomes than in a no-choice architecture context [47]. Thus, social media and other online applications can exploit digital nudging to play a leading role, such as against the sharing of fake news [67] and the protection of user privacy [68]. Particularly in crisis events, local organizations can use digital nudging to disseminate topic-specific tweets (e.g., emergency notifications, evacuation information, etc.) to support emergency management objectives and to manage the crisis properly [69]. In the long run, social media has become the most widespread channel for users to generate, access, and share all kinds of information. It is worthwhile to further explore how to use interface design and nudge to better assist users to search and share useful information more efficiently while privacy is effectively protected, and to guide other benign usage behaviors.

The blue cluster focuses on individual behavior, preferences, and the specific nudge mechanisms used in the implementation of behavioral interventions. At the early stage in applying Nudge Theory, Thaler and Sunstein [70] acronymized six mechanisms for optimizing choice systems to improve usage satisfaction into *nudge*, i.e., iNcentives, Understand mappings, Default, Give feedback, Expect error, and Structure complex choices. Since then, researchers have extended their design insights into additional fields, focused on the measurement and examination of influences that optimize nudge effects. As the network visualization shows, the main common nudge mechanisms are default options [71,72], social norms [73], incentives [74], and feedback [75]. Especially during the COVID-19 pandemic, nudge was used to promote vaccination, reduce social contact, disseminate trustworthy pandemic information, etc. [76]. In summary, the designer of any choice environment must be aware of its effect on people’s choices. Choice architects should be aware of the goals of the intervention to design and test nudge to maximize the desired effect [77]. Another study indicated that the vividness of image presentation increased gamification and improved the subjective usability of face-to-face counseling effects, which promoted counseling in real life for young people [78].

Finally, the yellow cluster, which concentrates on the extension of choice architecture to consumer behaviors that intervene in people’s consumption, such as healthy food choices, sustainable consumption behaviors, chronic disease or cancer prevention and treatment, and pro-environmental behaviors. The ever-accelerating pace of life, irrational eating patterns, and obsession with digital media affect physical and mental health, especially among students and young people in the workplace. Because of health and pro-social factors, most people are more receptive to nudge [79], although some studies have found that nudge to promote healthy eating is not effective [80], or even that employees can accept nudge while students do not accept nudge, leading to more unhealthy food choices and making nudge ineffective [81]. This may be because the nudge mechanism used is different and some nudges may be perceived as manipulative or uncomfortable, so the intervention is not as ideal [82]. In the context of disease prevention and treatment, it was demonstrated that moderate interventions in individual rights and relatively unproblematic moral imperatives, nudge proved valid in various situations [83], and in particular, that personally tailored and positively constructed messages were more persuasive than generic and/or negative messages [84].

On top of the network visualization, VOSviewer offered different mapping visualizations; the other two constructed are the overlay visualization (as in Figure 9) and the density visualization (as in Figure 10). Specifically, there are two groups of keywords in Figure 9 with the earliest average years of nudge research. The first group is Nudge Theory-related terms and research, such as “neoliberalism”, “behavioral economic”, “libertarian paternalism”, and “autonomy”. The second group is related to public health, such as “healthy food”, “disease”, and “obesity”, which is in line with the number of journal articles and citations discussed earlier and is a topic that has been studied early and consistently in nudge research. The above keywords are directly linked to both “nudge” and “choice architecture”, while several keywords on the right of the figure, such as “stimulation”, “sensitivity”, and “precipitation”, are linked to each other, but only “sensitivity” is linked to “nudge”. As there is no other connection that indicates relevance to the nudge discussed in this study, none are further reviewed. After 2019, increasing attention is paid to nudge in the digital environment, such as the keywords “digital nudging”, “fake news”, and “social media”, especially since the outbreak of COIVD-19 in early 2020. The spread of the pandemic has led to the implementation of stay-away orders and social distancing in many countries, in which personal use of social media multiplies. Researchers have also turned to the effects of nudge in this special period, with the keywords “COVID-19”, “vaccination”, and “hand hygiene” surging after 2020. Remarkably, there are also emerging keywords such as “sustainable consumption”, “artificial intelligence”, and “gamification”, indicating a growing enthusiasm for nudge research that extends to a growing number of fields.

The depth of research in the field related to nudge can be observed in Figure 10. In the present analysis, colors range from blue to green to yellow to red. The higher the number of items in the proximity of a point and the higher the weights of the related items, the closer the color of the point is to red. Contrarily, the sparser and less impactful the point, the closer its color to blue. Through density visualization, we can quickly observe that features around nudge research consumption, health, management, ethics, donation, sustainability, specific nudge mechanisms, and information systems are currently widely discussed topics.

## 4. Conclusions

It has merely been 14 years since the proposed Theory of Nudge, and there have been an increasing number of scholars around the world trying to find solutions to social issues by following the principles of nudge. This study provides a comprehensive view of the current trend in nudge studies. Methodologically, this study used a bibliometric analysis of nudge to perform analysis of journals, publication types, countries, authors, and keyword co-occurrence in a repository of 1706 publications retrieved from Web of Science. The results show that nudge studies have been extended to several disciplines beyond the concept of behavioral economics, and that nudge intervenes as a mediator in the relationship between humans and the world. The process of nudge implementation influences human perceptions and actions, experiences, and practices, which in turn influence the decision making of human behaviors. In the study, the results of the analysis laid the groundwork for answering the research questions presented in the beginning sections.

Specifically, (1) the number of nudge publications has increased significantly since 2012, especially after 2018, with the number of articles accounting for more than half of the total, and this is expected to keep growing. At present, nudge studies were first and most frequently conducted by American scholars, followed by European countries such as England, Germany, and The Netherlands, while contributions are lacking in other countries. (2) Economics is the most studied discipline of nudge, but with the adoption of multidisciplinary perspectives, political science, ethics, medical science, business, and communications have also become involved in exploring the application of nudge. This is particularly significant in the areas of public administration, healthcare, and sustainable consumption, in line with nudge’s goal of enabling the public to make the best choices about their health, wealth, and well-being. (3) Nudge’s future research potential and development direction are likely to continue to focus on people’s everyday behaviors, such as healthy lifestyles, sustainable consumption, and travel. Especially in the digital environment, there is a stronger need to help increase the well-being of the people being nudged through the design of the interface and the choice of decisions. Certainly, it is also important to be aware of the transparency and ethics of nudge in the design and application process to prevent the abuse of technology.

Finally, several limitations were faced by this study, which set out to expand our understanding of nudge from a macroscopic view. First, this analysis is limited to the data of publications retrieved from WoS. Though WoS is considered to primarily comprise accredited data sources, we cannot claim that the data are error-free and cover all studies. Therefore, we recommend that future researchers consider additional databases (e.g., Scopus, SpringerLink, EI, etc.) to ensure that important journal indexes and other time periods are covered. Second, the monolingual nature of the publications analyzed limited the inclusiveness and comprehensiveness of the present study. Only English-language publications were selected for the analysis, but in practice, it must be taken into account that most non-English publications are not included in the WoS. This is the case for 12 papers published in the journal *Acta Psychologica Sinica* in 2018, which provided exploratory research on the application of nudge in China from different perspectives in pro-environmental and pro-social areas—their absence in the collected data caused lacunae for the present study.

## Figures and Tables

**Figure 1 behavsci-13-00019-f001:**
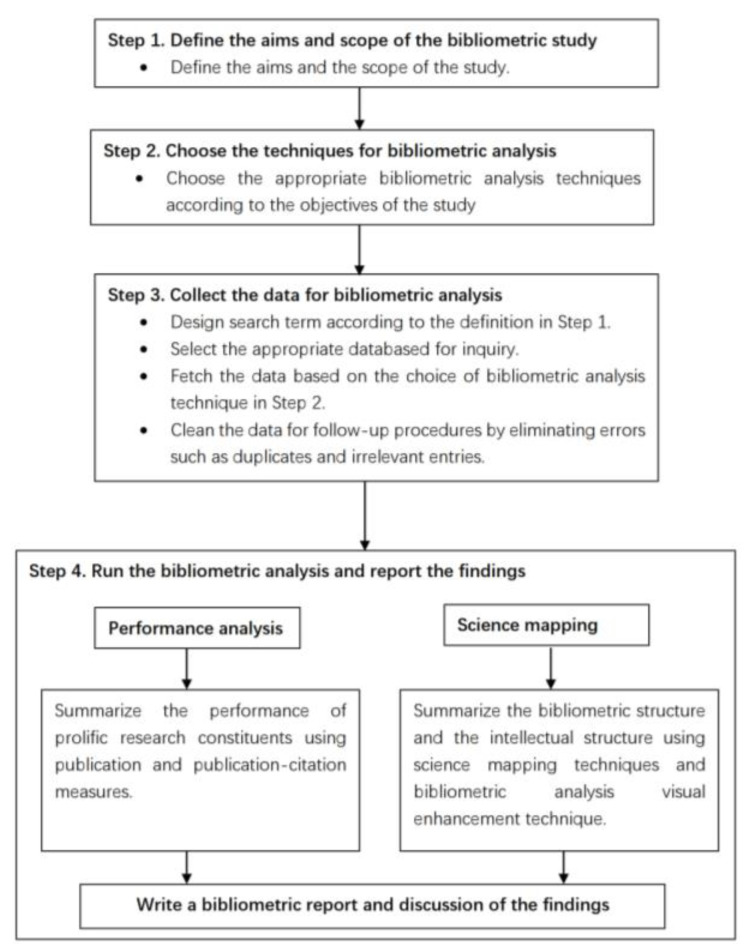
The bibliometric flowchart of this study.

**Figure 2 behavsci-13-00019-f002:**
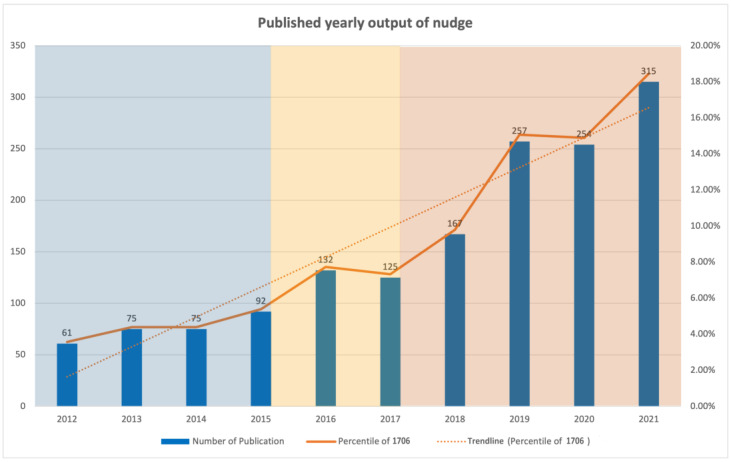
Published yearly output of nudge.

**Figure 3 behavsci-13-00019-f003:**
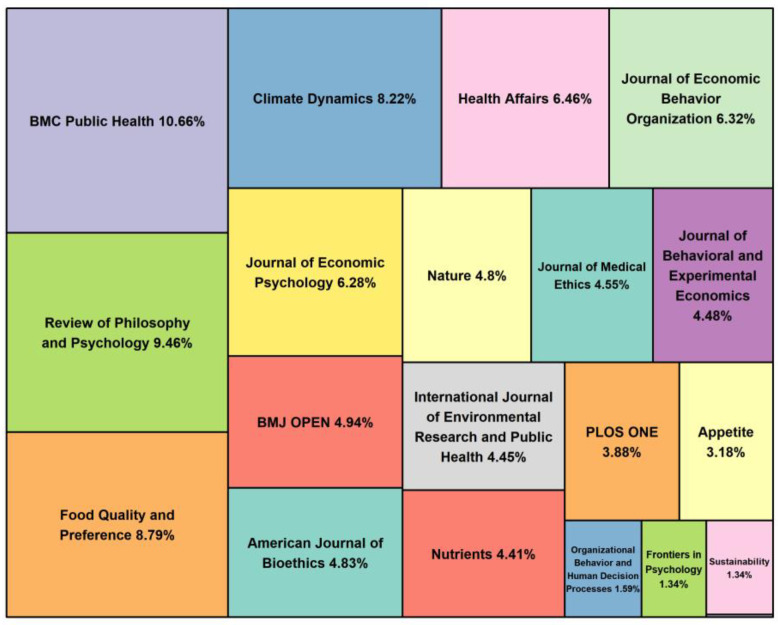
Treemap of the top 20 journals with average citations.

**Figure 4 behavsci-13-00019-f004:**
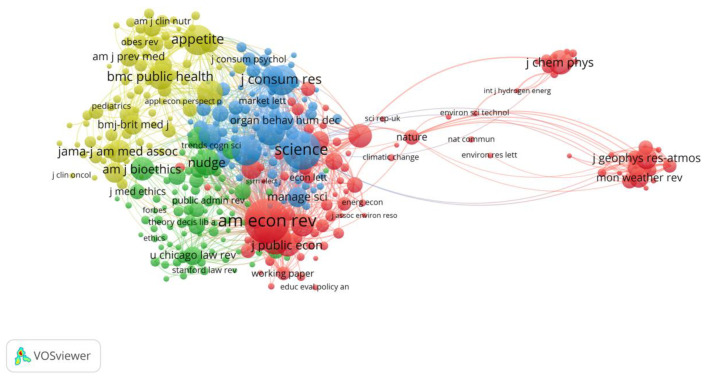
The co-citation network of journals.

**Figure 5 behavsci-13-00019-f005:**
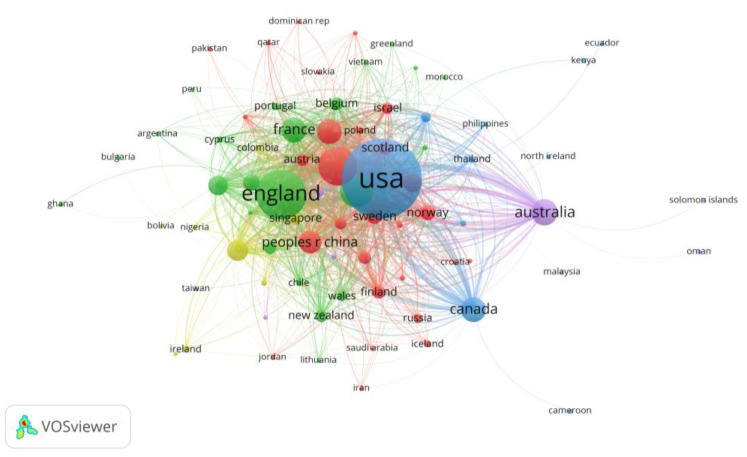
The bibliographic coupling network visualization of countries.

**Figure 6 behavsci-13-00019-f006:**
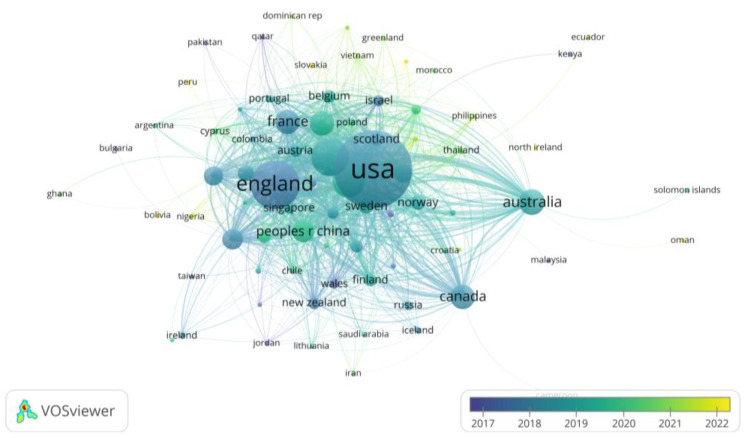
The overlay visualization of the country analysis.

**Figure 7 behavsci-13-00019-f007:**
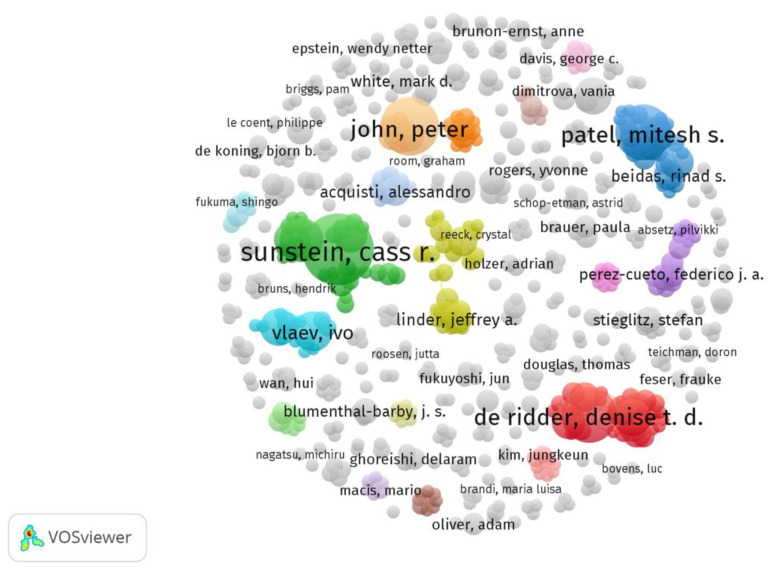
Network visualization for co-authorship analysis.

**Figure 8 behavsci-13-00019-f008:**
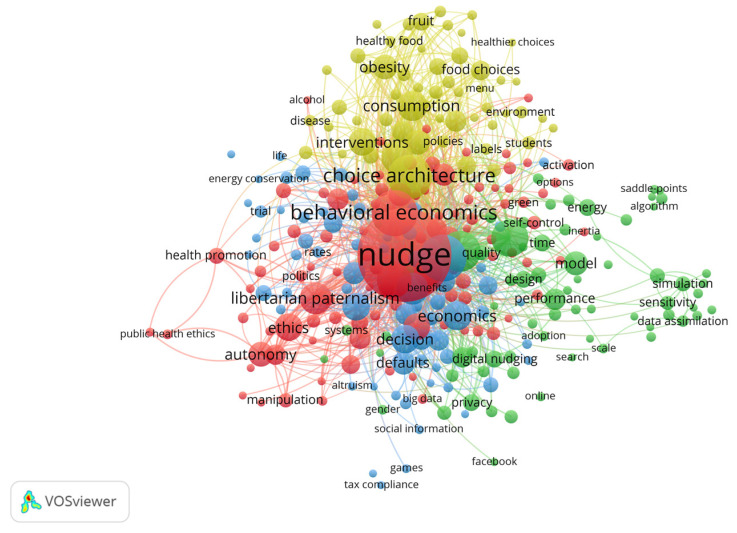
Network visualization of keyword co-occurrence analysis.

**Figure 9 behavsci-13-00019-f009:**
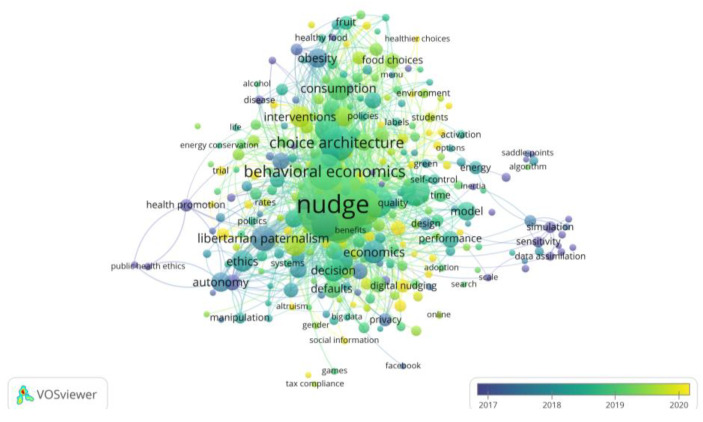
Overlay visualization of keyword co-occurrence analysis.

**Figure 10 behavsci-13-00019-f010:**
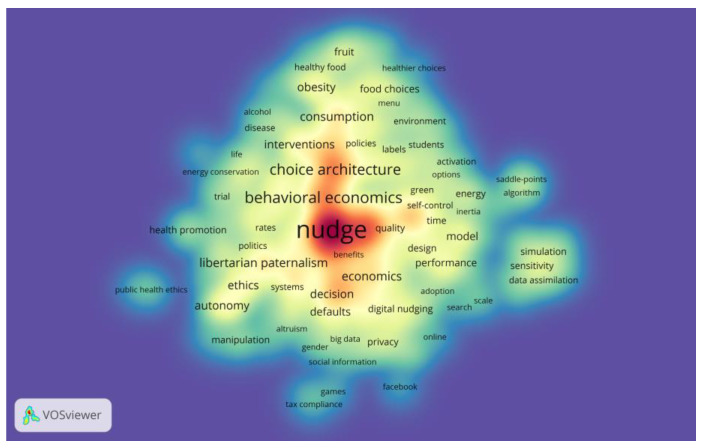
Density visualization of keyword co-occurrence analysis.

**Table 1 behavsci-13-00019-t001:** Publications by document type.

Type of Publication	Number of Publications	Percentage of Total Publications (%)
Article	1217	71.34%
Editorial Material	145	8.50%
Conference Paper	128	7.50%
Review	65	3.81%
Others	151	8.85%
Total	1706	100%

**Table 2 behavsci-13-00019-t002:** Top 20 research categories of nudge publications.

Web of Science Categories	Number of Publications	Percentage of Total Publications (%)
Economics	226	13.25%
Public Environmental Occupational Health	116	6.80%
Ethics	102	5.98%
Psychology Multidisciplinary	94	5.51%
Political Science	78	4.57%
Public Administration	77	4.51%
Law	74	4.34%
Social Sciences Biomedical	72	4.22%
Multidisciplinary Sciences	70	4.10%
Business	69	4.05%
Environmental Sciences	69	4.05%
Nutrition Dietetics	69	4.05%
Computer Science Theory Methods	67	3.93%
Health Care Sciences Services	67	3.93%
Environmental Studies	65	3.81%
Social Issues	65	3.81%
Medical Ethics	63	3.69%
Management	60	3.52%
Computer Science Information Systems	59	3.46%
Meteorology Atmospheric Sciences	52	3.05%

**Table 3 behavsci-13-00019-t003:** Top 20 most productive journals.

Journal	Publications	Citation	AC	Index	IF
*American Journal of Bioethics*	33	453	13.7	SSCI/SCIE	14.676
*PLOS ONE*	21	237	11	SCIE	3.752
*Appetite*	20	180	9	SCIE	5.016
*Frontiers in Psychology*	20	76	3.8	SSCI	4.232
*Journal of Medical Ethics*	17	220	12.9	SSCI/SCIE	5.926
*BMC Public Health*	16	483	30.2	SCIE	4.135
*Journal of Economic Behavior Organization*	15	269	17.9	SSCI	2
*New Scientist*	14	2	0.1	SCIE	0.319
*Food Quality and Preference*	13	324	24.9	SCIE	6.345
*Organizational Behavior and Human Decision Processes*	12	54	4.5	SSCI	5.606
*Sustainability*	11	42	3.8	SSCI/SCI	3.889
*Review of Philosophy and Psychology*	11	295	26.8	ESCI	1.337
*Journal of Behavioral and Experimental Economics*	10	127	12.7	SSCI	1.831
*BMJ OPEN*	9	126	14	SCIE	3.006
*Nature*	9	122	13.6	SCIE	69.504
*Climate Dynamics*	8	186	23.3	SCIE	4.901
*Health Affairs*	8	146	18.3	SSCI/SCIE	6.301
*International Journal of Environmental Research and Public Health*	8	101	12.6	SSCI/SCIE	4.614
*Journal of Economic Psychology*	8	142	17.8	SSCI	3
*Nutrients*	8	100	12.5	SCIE	6.706

AC = average citations; SCIE/SSCI = Science Citation Index Expanded/Social Science Citation Index; IF = impact factor.

**Table 4 behavsci-13-00019-t004:** Top 10 publication countries (n = 1548).

Country	Publications	Citations	Link Strength
USA	631	10,730	2813
England	260	3418	1670
Germany	159	1986	1648
The Netherlands	116	1431	1350
Denmark	54	1433	1116
Australia	79	867	693
Italy	71	511	601
Sweden	32	865	549
Canada	73	1377	545
France	73	915	541

**Table 5 behavsci-13-00019-t005:** Prominent authors by documents and citations.

Author	Documents	Citations	H-Index
Cass R. Sunstein	20	1057	84
Peter John	15	23	66
Mitesh S. Patel	14	197	34
Denise de Ridder	14	257	49
Anne Thorndike	10	465	30
Ivo Vlaev	9	105	24
Douglas Levy	8	445	27
Bart Engelen	8	127	9
Magda Osman	8	74	19
Joline Beulens	8	58	62

## Data Availability

Not applicable.

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
