# Peer review of "A Bibliometric Analysis and Review of Nudge Research Using VOSviewer"

_behavsci, 2022, doi:10.3390/bs13010019_

Round 1

Reviewer 1 Report

A Bibliometric Analysis and Review of Nudge Research Using VOSviewer

by Chenjin Jia , Hasrina Mustafa   

General comments

The manuscript is quite well written, the research questions are valid, the methodology is sound even if it needs clarifications, the conclusions are supported by the results whose discussion needs to be deepened. The bibliography needs to be expanded. 

Specific points

Abstract. Line 16 "analyzing", not "studying".

1. Introduction. The authors will have to explain why they preferred the bibliometric study to a traditional review work.

At the end of the section, the authors will include one or two paragraphs explaining who and why can benefit of the research results. In additions, the authors should inform the readers how the manuscript is organized (its structure). Therefore, an additional paragraph is suitable.

2. Materials and Methods   The authors will better explain what the general goals of bibliometric studies are. In doing so, it is required to discuss proper literature.

The description of the methodology is a bit confusing. The authors first introduce the software used (VOSviewer) and then discuss where the data implemented in the software was obtained. Reverse the order.   The authors used Web of Science for bibliographic data search. However, it is required to justify the choice. Why didn't the authors consider other databases (e.g., Scopus)? Was a comparison made between the different databases? Which deductions can be made from the comparisons?  Furthermore, the bibliometric study requires a systematic protocol that allows the study to be reproducible and reliable. This is essential to make the research sound. Therefore, the authors will clarify all the criteria considered. This phase is mandatory for bibliometric studies. Please include a flow chart able to clarify the whole path followed for the study.It is suitable to highlight the well-extablished use of VOSviewer by addingrecent literature:  doi.org/10.3390/geosciences10120482 doi.org/10.3390/app10062080  doi.org/10.3390/en14010162

3. Results. The section should be renamed "Results and Discussion" as the results of the bibliometric analysis are discussed in relationship to the previous studies.

The graph of Figure 1 should highlight and recall the subdivision into the three time phases described in the text. Therefore, the authors will include three vertical lines in the graph, delimiting the three phases discussed in the manuscript (lines 185-195).

3.1.1. Journal distribution

Lines 246-247. Which can be the reasons for such high citations? Can we highlight which research has received so much attention from the scientific community?

Line 269. Indicate how many nodes there are for each cluster.

3.1.2. Country and institution distribution

LInes 329-330.  In order to highlight which countries have recently begun to be active in nudge research, the authors should include the 'overlay map' related to the bibliographic coupling between countries.

3.1.3. Author distribution

Line 388-390. This statement needs to be clarified and better linked to the data. I'd rather not read terms like "superstar" in the manuscript.

3.1.4. Keywords co-occurrence analysis

Line 483, "Figure 7 has..." should be "Figure 6 has..."

Figure 6 is not discussed. Therefore, the author will analyze the "historical" keywords and those that have emerged in the recent years.

Concluding, I encourage the authors to revise and submit the new version of the manuscript.

KInd regards

Author Response

Dear Reviewer #1,

Greetings,

Thanks for your efforts and time reviewing our manuscript. After revision and discussion, we found the overall quality of the manuscript has dramatically improved. Please find the attached file for the responses to your comments and suggestions.

Reviewer 2 Report

Dear Authors, 

Reading a review paper about nudge was entertaining. It is a timely and fresh topic. However, there are several issues to be addressed. 

1. I expect to see something more than simple description of the summary of 1706 papers or alternatively provide reasons why it needs to be explained. Now, the paper describes the number of publications, journals, countries, authors, and keywords without any reason. In this case, countries (3.1.2.) does not seem to be as valuable as other components. 

2. It needs to add recent publications of the Behavioral Science. One example is Piao and Joo (2022) in which gamification as a nudge helped young people to adopt in-person counseling. 

3. Incomplete sentences and typos need to be corrected. For instance, Line 50 is an incomplete sentence with a question mark in the middle. Thaler was misspelled in line 212. 

I look forward to seeing a revised manuscript which is more tighten-up, contains more fresh findings, and is flawless. 

Author Response

Dear Reviewer #2,

Greetings,

Thanks for your efforts and time reviewing our manuscript. After revision and discussion, we found the overall quality of the manuscript has dramatically improved. Please find the attached file for the responses to your comments and suggestions. 

Round 2

Reviewer 1 Report

The authors performed a thorough revision addressing all the issues raised by the reviewer. The manuscript is now clear both in the methodology and in the discussion of the results.

I only recommend improving the quality of Figure 3: the text included in the figure is not readible. After this correction, in my opinion, the manuscript can  be published.

Kind regards

Author Response

Dear Reviewer 1,

Thank you for your patient review and comments.

We have recreated Figure 3 to make it clearer and more readable.

Reviewer 2 Report

Authors successfully revised manuscript following reviewer's comments. 

Author Response

Dear Reviewer 2,

Thank you for your patient review and comments.